# Root Cause Analysis of Patients with Pancreatic Cancer Who Underwent Imaging Not Resulting in a Cancer Diagnosis in the 18 Months Prior to Diagnosis

**DOI:** 10.3390/cancers17233770

**Published:** 2025-11-26

**Authors:** Shahd A. Mohamed, Nosheen Umar, Melisa Sia, Justin Smyth, Sumedha Udumalagala, Mujeeb Ullahj Makki, Keith Roberts, Brinder Mahon, Raneem Albazaz, Nigel Trudgill

**Affiliations:** 1Department of Gastroenterology, Sandwell and West Birmingham NHS Trust, Hallam Street, West Bromwich B71 4HJ, UK; shahdahmed@nhs.net (S.A.M.); nosheen.umar@nhs.net (N.U.); m28887@pgim.cmb.ac.lk (S.U.); mujeeb.makki2@nhs.net (M.U.M.); 2Leeds Teaching Hospitals NHS Trust, Great George Street, Leeds LS1 3EX, UK; melisa.sia@nhs.net (M.S.); r.albazaz@nhs.net (R.A.); 3University Hospital Birmingham NHS Foundation Trust, Mindelsohn Way, Edgbaston, Birmingham B15 2WB, UKkeith.roberts@uhb.nhs.uk (K.R.);; 4Department of Cancer and Genomic Sciences, Robert Aitken Building, University of Birmingham, Vincent Drive, Edgbaston, Birmingham B15 2TT, UK

**Keywords:** pancreatic cancer, early diagnosis, imaging

## Abstract

For the first time, the present study examined the imaging and clinical care of patients with pancreatic cancer not diagnosed on index imaging (post-imaging pancreatic cancer—PIPC) and developed a root cause analysis of the most likely explanations for it. A total of 35% of PIPCs were classified as potentially avoidable. Twelve (26%) included a missed focal lesion; four (9%) had a focal lesion or duct dilatation that was inadequately or not investigated. This root cause analysis tool can support providers in conducting their own reviews of potentially missed pancreatic cancer cases by systemically identifying diagnostic, technical or pathway-related factors. This can inform targeted strategies for earlier detection and improved patient outcomes.

## 1. Introduction

Global trends suggest that pancreatic ductal adenocarcinoma incidence rates are increasing, and it is expected to become the second leading cause of cancer deaths in the West [1]. Patients with pancreatic cancer commonly present with non-specific symptoms during the early course of the disease [2], which may result in delayed diagnoses [3,4,5,6,7]. Surgical resection provides the only curative therapy, but presenting with potentially resectable disease is uncommon (only 15–20% of pancreatic cancer patients) [1,2,8,9,10]. So, it is essential to try to detect pancreatic cancer at an early stage, including opportunistically during abdominal imaging, in order to improve patient outcomes.

Although CT scans are the most commonly used imaging modality for diagnosing pancreatic cancers [11,12,13], the diagnosis can still be missed on these scans. A meta-analysis highlighted that CT has a sensitivity and specificity of only 89% and 90%, respectively, for pancreatic cancer [14]. A recent study comparing the diagnostic accuracy of convolutional neural networks and radiologists reported that 7% of pancreatic cancers were missed by radiologists on CT [15]. Failure to diagnose pancreatic cancer on cross-sectional imaging can lead to missed opportunities to treat the disease at an earlier stage, as there is often only a short time window for curative surgery for pancreatic cancer [16]. Previous studies have reported pancreatic cancer to be present in 83% [17] and 63–100% [18] of CT scans, respectively, 0–18 months before diagnosis. Patients with a delay in diagnosis of pancreatic cancer were significantly more likely to have the following features on their pre-diagnostic scan: focal hypoattenuation, pancreatic duct dilatation, interruption of the pancreatic duct and distal parenchymal atrophy [17,18]. While another study found 25.7% of scans had a mean imaging misdiagnosis interval of 8.1 months (RADPEER 2) and 5 months (RADPEER 3), respectively, for pancreatic cancer [19]. Factors that may have confounded the detection or interpretation of pancreatic cancer on index imaging were reported to be: inherent tumour factors (tumours less than 2 cm, iso-attenuating or non-contour deforming), secondary signs (e.g., vascular encasement, pancreatic duct dilatation or cut-off), concurrent pancreatic pathology (e.g., pancreatitis), technical limitations (e.g., non-pancreatic protocol, lack of diffusion-weighted imaging on MRI, intravenous contrast dosing) and cognitive biases (e.g., inattentional blindness) [20]. 

Post-colonoscopy colorectal cancer (PCCRC), post-endoscopy upper gastrointestinal cancer (PEUGIC) and post-imaging (CT colonography) colorectal cancer (PICC) are defined as cancers diagnosed within 6 to 36 months of an endoscopy or CT scan which did not find a cancer. They are recommended as key performance indicators to assess and compare the performance of endoscopy and radiology providers, respectively, in the UK [21,22,23,24]. The World Endoscopy Organisation has proposed a root cause analysis system to identify potentially avoidable PCCRC and PICC and their most plausible explanations [25]. For the first time, the present study examined imaging and clinical care of patients with pancreatic cancer not diagnosed on index imaging (post-imaging pancreatic cancer—PIPC) and developed a root cause analysis of the most plausible explanations for it. Although pancreatic cancer has a low prevalence, it is one of the six least survivable cancers in the UK [26], with no screening modality, unlike colorectal cancer. This root cause analysis tool can support providers to conduct their own reviews of potentially missed cases. By systematically identifying diagnostic, technical or pathway-related factors, the tool can inform targeted strategies to help with its earlier detection and improve patient outcomes.

## 2. Methods

### 2.1. Study Subjects

This was a retrospective study, in which records of patients aged 18 and above coded with pancreatic cancer were identified at two UK hospitals: Sandwell and West Birmingham NHS Trust (January 2016 to August 2021) and Leeds Teaching Hospitals NHS Trust (January 2019 to August 2021). Patients with a diagnosis of cancer other than pancreatic cancer, benign pancreatic pathology (adenoma, cystadenoma, pseudocyst), and intraductal papillary mucinous neoplasm, who had been miscoded, were excluded following review of clinical records.

PIPC was defined as pancreatic cancer diagnosed 3 to 18 months after an index abdominal cross-sectional (CT or MRI) imaging that did not diagnose pancreatic cancer (Figure 1 and Figure 2). This was based on the methodology of previous studies of PIPC [17,18,19]. The index imaging that did not diagnose the pancreatic cancer and the diagnostic scan were reviewed by two hepatobiliary radiologists. Each radiologist reviewed scans in one provider and determined their PIPC category. Subsequently, each radiologist reviewed the other provider’s imaging and the assigned PIPC category in order to note any discrepancy. The radiologists also assessed whether the imaging was adequate to exclude pancreatic cancer, if undertaken for a different purpose (e.g., CT renogram). 

Variables collected included demographics (age, sex), indication for imaging divided into abdominal alarm symptoms (obstructive jaundice, abdominal pain and weight loss, anaemia, change in bowel habit to diarrhoea), other abdominal symptoms (dyspepsia, abdominal pain without weight loss, nausea/vomiting, other change in bowel habit), other indication for imaging, source and urgency of scan (outpatient routine/urgent/surveillance, inpatient urgent), imaging modality (CT/MRI), reporting radiologist’s subspecialty (categorised into gastrointestinal or hepatobiliary radiologist or non-gastrointestinal or hepatobiliary radiologist).

### 2.2. Root Cause Analysis of the Most Likely Explanation for PIPC

Following review of index and diagnostic imaging and the clinical records of patients, PIPCs were categorised into the following types and subtypes.

Type 1: Focal lesion on index scan reported in the same pancreatic segment as diagnostic scan.

If yes, categorise into the following subtypes, if no, proceed to Type 2:Further assessment plan adequate and within appropriate time frame, but still PIPC.Further assessment plan inadequate.Further assessment not undertaken or not undertaken within correct time frame, but appropriate due to patient choice or comorbidity.Further assessment not undertaken/within an appropriate (urgent—less than two weeks) time frame, and inappropriate.

Type 2: Imaging changes that can be associated with pancreatic cancer (e.g., bile or pancreatic duct dilatation) reported on index scan.

If yes, categorise into the following subtypes, if no, proceed to Type 3:Further assessment plan adequate and within appropriate time frame but still PIPC.Further assessment plan inadequate.Further assessment not undertaken or not undertaken within correct time frame but appropriate due to patient choice or comorbidity.Further assessment not undertaken/within appropriate time frame and inappropriate.

Type 3: Missed focal lesion or imaging changes associated with pancreatic cancer on index scan.

If yes, categorise into the following subtypes, if no, proceed to Type 4:Focal lesion in same pancreatic segment on later review only.Imaging changes (e.g., bile or pancreatic duct dilatation) that can be associated with pancreatic cancer noted on later review only.

Type 4: No focal lesion or imaging changes associated with pancreatic cancer on index scan.

If yes, categorise into the following subtypes:No lesion/imaging changes, and imaging adequate to exclude pancreatic cancer.Possible missed lesion, prior imaging inadequate to exclude pancreatic cancer.

### 2.3. Judgement on Whether PIPC Was Potentially Avoidable

Whether PIPCs were potentially avoidable was assessed using the root cause analysis. Pancreatic cancers not associated with a lesion or imaging changes associated with pancreatic cancer on adequate index imaging, or if index imaging was inadequate to rule out pancreatic cancer, were both deemed unavoidable. PIPCs were also considered unavoidable if the recommended pathway was not followed due to a patient’s refusal of further investigations or if the patient was too frail for further investigations. Finally, if the further follow-up plan was adequate to investigate imaging changes associated with pancreatic cancer and within an appropriate time frame, then such PIPCs were also considered unavoidable.

### 2.4. Potential Impact of a Delay in Diagnosis on PIPC Clinical Outcomes

The outcome of patients with a missed focal lesion or patients with an inadequate further management plan after identifying imaging signs associated with pancreatic cancer on their prior imaging could potentially be different if they were diagnosed at an earlier stage. The outcomes of patients who were frail at index imaging and were unlikely to be eligible for curative treatment at any stage were considered unlikely to be different. Patients diagnosed at an advanced stage with inadequate index imaging to rule out pancreatic cancer were also considered unlikely to have a different outcome.

The scans of patients with potentially missed focal pancreatic lesions were subsequently reviewed by a hepatobiliary surgeon to assess whether lesions were potentially resectable on both index and diagnostic imaging.

### 2.5. Statistical Analysis

Continuous variables are presented as mean (SD) and median (IQR). Categorical variables are presented as percentages. Rank sum and Chi-squared tests were used for continuous and categorical variables, respectively, and two-sided *p* values were considered significant at <0.05. Stata statistical software release 17 was used for statistical analysis.

### 2.6. Patient and Public Involvement

A patient and public participation event was held with the support of Pancreatic Cancer UK to discuss the design and delivery of the PIPC root cause analysis project. One patient had experienced PIPC. Improving the efficiency of diagnosis and increasing the public confidence in investigations, such as scans were seen as key contributors to patients presenting promptly for investigation of symptoms.

## 3. Results

### 3.1. Study Patients

Following the exclusion of 19 patients without pancreatic cancer, the clinical records and imaging of 600 patients were reviewed (Figure 3). A total of 46 of 600 patients (7.7%, 95% CI 5.7–10.1%) were classified as having PIPC, with 43 CT and 3 MRI undertaken 3–18 months before cancer diagnosis. Median age 75.9 (IQR 69.6–80.2) years, and 58.7% female. The demographic details of patients with PIPCs are shown in Table 1. Detailed data were available for all patients with PIPCs, but were limited in other pancreatic cancer patients without PIPC to patients at Sandwell and West Birmingham NHS Trust. Data from pancreatic cancer patients with and without PIPC are shown in Table 2.

For all PIPC patients, the median time interval between the diagnostic scan and index imaging was 11 months (IQR 7.4–14.9). In total, 39.1% of PIPC patients were diagnosed with pancreatic cancer based on imaging alone, and 60.9% had a histological diagnosis, compared with 66.9% and 33.1%, respectively, of pancreatic cancer patients without PIPC (*p* = 0.01). Approximately 75% of scans in both groups were reported by non-gastrointestinal or hepatobiliary radiologists.

### 3.2. Root Cause Analysis of the Most Likely Explanation for PIPC

There were no disagreements between the two radiologists regarding the categorisation of PIPC. The results of root cause analysis of PIPC patients are shown in Figure 4. PIPC patients were classified into the four subtypes: Type 1–1 (2.2%); Type 2–8 (17.4%); Type 3–12 (26.1%); and Type 4–25 (54.3%).

In the eight patients with pancreatic or common bile ductal dilatation on diagnostic imaging, electronic records were reviewed to assess whether the duct dilatation was present on imaging more than 18 months before pancreatic cancer diagnosis. Five patients had no previous imaging over 18 months before diagnosis. Three patients had imaging more than 18 months before diagnosis, and the duct dilatation on their PIPC imaging was a new finding in each patient.

Eight patients (66.7%) with a missed focal lesion on the PIPC imaging (Type 3 PIPC) had alarm symptoms, including obstructive jaundice, abdominal pain and weight loss, or a change in bowel habit to diarrhoea, prompting their PIPC imaging.

### 3.3. Potentially Avoidable PIPC

Sixteen (35%, 95% CI 21.4–50.2%) PIPCs were classified as potentially avoidable: twelve (26%) Type 3 with a missed focal lesion; four (9%) with a focal lesion or duct dilatation inadequately or not investigated. Unavoidable PIPC included one patient who refused any further investigation and one patient in whom the decision was made not to investigate further due to multiple co-morbidities and frailty.

### 3.4. Potential Impact on Pancreatic Cancer Clinical Outcomes

On review, ten (83.3%, 95% CI 51.6–97.9%) patients with Type 3 PIPC (a missed focal lesion) had a potentially resectable lesion on their PIPC imaging, but only four (33.3%, 95% CI 9.9–65.1%) were still potentially resectable on their diagnostic imaging.

The clinical staging on the diagnostic scan and the treatment received by all pancreatic cancer patients are presented in Table 3. Seventy eight percent of PIPCs were later diagnosed at an advanced stage (stage III and IV) and 68.5% of pancreatic cancer patients without PIPC (*p* = 0.90). No significant differences were observed in the treatment approaches between pancreatic cancer patients with and without PIPC.

## 4. Discussion

In this study, the proportion of potentially missed pancreatic cancers on imaging at two healthcare providers was assessed and a new root cause analysis approach developed to establish the most plausible explanation. At a PIPC rate of 7.7%, this study suggests that of the 10,452 new patients with pancreatic cancer annually in the UK, around 800 patients may have PIPC each year and 280 are potentially preventable.

This study is the first to systemically examine the imaging and clinical management of patients with PIPC who had focal or imaging abnormalities such as duct dilatation, that can be associated with pancreatic cancer. In 19.6% of patients, there was a focal abnormality or duct dilatation reported on the index scan, but it was not investigated, although in 1 of 9 patients this was appropriate due to patient frailty. In 26.1% there was a missed focal lesion on the index imaging. In 54.3% there was no concern on index imaging or subsequent management. Overall, 35% of PIPCs were deemed potentially avoidable, with more than three-quarters considered potentially resectable at the time of PIPC imaging if they had been diagnosed at the time. 

In contrast, a systematic review and meta-analysis of 19,867 patients found the post-imaging colorectal cancer (PICC) rate within 36 months of initial imaging to be 4.42%. However, they reported that in 61% of cases, the culprit lesion was visible in retrospect and potentially detectable, while errors in patient management were less common [24]. However, important differences exist between PICC and PIPC. CT colonography is typically interpreted by gastrointestinal radiologists to exclude polyps greater than 1 cm or colorectal cancers. In contrast, PIPC scans are more heterogeneous, encompassing CT or MRI performed for a variety of clinical indications and are often interpreted by non-gastrointestinal radiologists (67.4% in this study). Given that approximately 65% of PIPC cases were deemed not preventable in the present study, overall PIPC rates have limited value as a key performance indicator. Local root cause analyses of PIPC and interventions to reduce preventable PIPC in the future, using the methodology in this study, will hopefully reduce preventable PIPC in the future. 

A previous study reported the proportion of missed or misinterpreted pancreatic cancers on imaging up to 5 years before diagnosis in 2014 and 2015 at a single healthcare provider in Canada [19,20]. Even though a different time period prior to diagnosis was examined and ultrasound examinations were also included, a very similar proportion of 25.7% of pancreatic cancer patients experienced a missed opportunity for earlier diagnosis, where a focal lesion was missed on imaging, compared with 26% in the present study. 

Discrepancies in radiological reporting are not uncommon, and abdominopelvic imaging has a higher rate of discordance than other areas due to the broad range of pathologies and its complex anatomy [27]. The sub-specialty of the reporting radiologist and whether they have a hepatobiliary or gastrointestinal sub-specialty or interest is likely to contribute to missed PIPC. In the current study of 46 PIPC, 12 were type 3a (an unrecognised focal lesion in the same pancreatic segment on later review), of which only 2 were reported by hepatobiliary/gastrointestinal radiologists, whereas 32.6% of PIPC scans were reported by hepatobiliary/gastrointestinal radiologists, although this difference fell short of statistical significance due to small numbers. Given the volume of abdominal imaging undertaken, including, for example, for urological or vascular indications, it will not be feasible for all such imaging to be reported by radiologists with a hepatobiliary or gastrointestinal subspecialty interest in the UK at least. 

It is worth noting that only nine (56%) of the potentially preventable PIPC had alarm features suggesting pancreatic cancer, such as weight loss and abdominal pain or jaundice. Other authors have noted that only 6.2% of PIPC patients had alarm symptoms [17], and the lack of such symptoms is likely to contribute to less focus on the pancreas as a potential source of the patients’ symptoms. The use of artificial intelligence may have a role in reducing the frequency of such missed PIPC [28]. Artificial intelligence has been reported to achieve on non-contrast CT scans an area under the receiver operating curve (AUC) of 0.986–0.996, with a sensitivity of 92.9% and specificity of 99.9% [28]. This includes an increase in sensitivity of 34.1% and specificity of 6.3% compared with radiologists.

In terms of the time interval from index to diagnostic scan, it was 11 months for PIPC patients in the current study, which is similar to three other studies where the mean time interval was 13.7 months, 13.3 months and 18 months, respectively [18,29,30]. The potential impact of the diagnostic delay on the outcome of treatment was also examined in the current study. We found that the average delay in the 16 patients with potentially preventable PIPC was 8.3 (6.6–11.9) months. For patients with Type 3 PIPC, a missed focal lesion on index imaging, 10 of the 12 patients were potentially resectable based on their imaging that failed to diagnose cancer. However, following the delays in diagnosis to their diagnostic pathway, only 3 (13%) of the PIPC patients underwent surgery following diagnosis. Others have reported similar findings with 25% potentially resectable on a CT scan within 6 months of diagnosis, but only 11.5% resectable at the time of their diagnostic scan [29].

## 5. Limitations

There are several limitations to the current study. This is a retrospective study introducing a bias in the radiologists reviewing the diagnostic and non-diagnostic imaging, given their awareness of a later pancreatic cancer diagnosis. The radiologists subsequently reviewed the PIPC reviewed by the other radiologist with no discrepancies noted in their assessments, but this expectation bias remains and may overestimate the number of subtle lesions on non-diagnostic imaging. It is important to note that this study involved only two healthcare providers, which makes the findings less generalisable, as local radiology reporting issues or pathways for the investigation of possible pancreatic malignancy at the two providers will contribute to the findings reported. Both providers are part of large conurbations, and some patients may have been diagnosed with PIPC at different providers and would not have been captured in an analysis limited to local hospital records, meaning the PIPC rate found is likely to be an under-estimate. However, in general, the findings were consistent with previous similar studies, reducing these possible concerns. It is also worth noting that only 45.7% of PIPC patients had a histologically confirmed diagnosis of PC. This will relate to the fact that many of these patients were finally diagnosed at an advanced stage and were unlikely to be candidates for chemotherapy or surgical treatments at that stage, but this is a limitation. Unfortunately, detailed information on pancreatic cancer patients without PIPC was limited to one of the providers, limiting the ability to compare PIPC patients and pancreatic cancer patients without PIPC. Further work in a larger number of providers is clearly needed to develop the findings from this study and we are currently setting up a national study in all hospitals in England.

## 6. Conclusions

In conclusion, approximately 8% of patients with pancreatic cancer have imaging that fails to diagnose their cancer in the 18 months prior to diagnosis. Root cause analysis highlighted that in a third, there was an unrecognised focal lesion in the pancreas or pancreatic and/or biliary duct dilatation that is associated with pancreatic cancer. These represent missed opportunities to diagnose pancreatic cancer at an earlier, potentially more treatable stage.

## Figures and Tables

**Figure 1 cancers-17-03770-f001:**
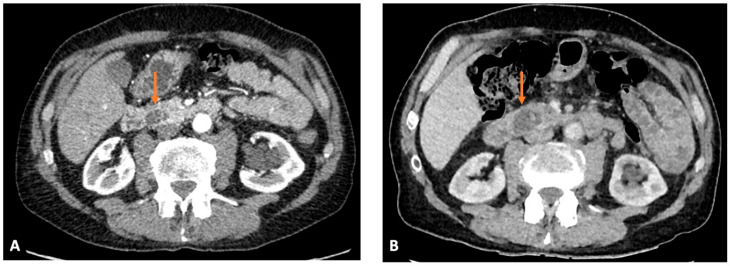
Case examples of Post Imaging Pancreatic Cancer (PIPC). Type 2a: Pancreatic cancer associated abnormality detected, with adequate subsequent management, but still PIPC. (**A**) Axial arterial phase and (**B**) subsequent portal venous phase CT images 18 months apart. Baseline CT 3 months post episode of pancreatitis shows a 2 cm lesion in the head of pancreas thought to be inflammatory, this was also subsequently investigated with EUS which showed no suspicious lesions. CT 18 months later for recurrent symptoms showed the lesion has enlarged, with new pancreatic duct dilatation. Repeat EUS with sampling demonstrated a mucinous adenocarcinoma.

**Figure 2 cancers-17-03770-f002:**
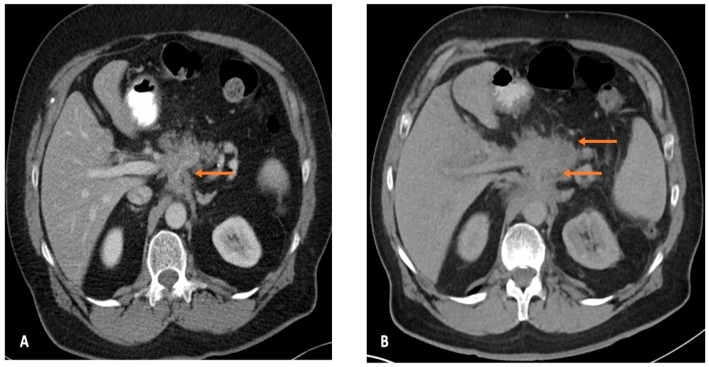
Case examples of Post Imaging Pancreatic Cancer (PIPC). Type 2b: Pancreatic cancer associated abnormality detected (e.g., pancreatic or bile duct dilatation but no focal lesion) but inadequate follow up plan. (**A**) Axial CT image in portal venous phase demonstrating abnormal soft tissue around the coeliac axis with loss of fat planes interpreted as retroperitoneal fibrosis. (**B**) Subsequent CT 7 months later shows significant increase in volume of abnormal soft tissue around the coeliac axis with now a solid pancreatic body mass replacing the pancreatic parenchyma with loss of normal pancreatic lobulations. Abnormality was identified but there was an interpretation error and, therefore, inadequate subsequent management.

**Figure 3 cancers-17-03770-f003:**
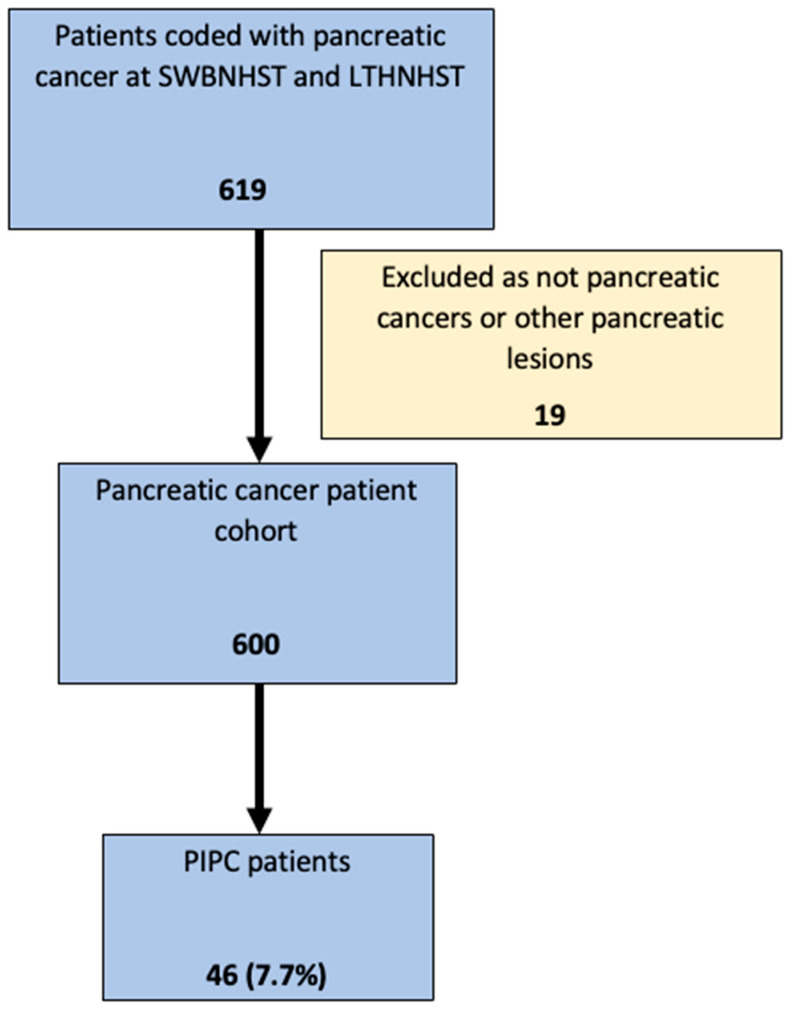
Study flow diagram. LTH NHST—Leeds Teaching Hospitals National Health Service Trust. SWBH NHST—Sandwell West Birmingham National Health Service Trust. PIPC—Post Imaging Pancreatic Cancer.

**Figure 4 cancers-17-03770-f004:**
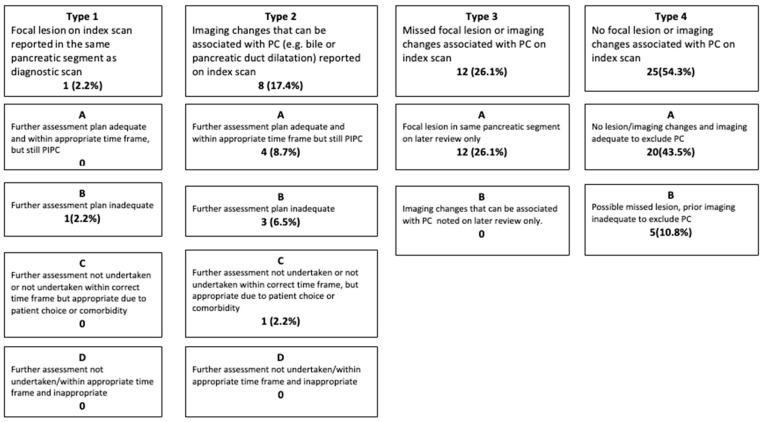
Root cause analysis for the most likely explanation for PIPC. PIPC—Post Imaging Pancreatic Cancer; PC—pancreatic cancer.

**Table 1 cancers-17-03770-t001:** Demographic details of patients with Post Imaging Pancreatic Cancer.

Variables		PIPC (*n* = 46)	%
**Time interval between scans (months)**		11 (IQR 7.4–14.9)	
**Age (years)**		75.9 (IQR 69.6–80.2)	
**Sex**	Female	27	58.7
	Male	19	41.3
**Location of pancreatic cancer**	Body	11	23.9%
	Body/Tail	5	10.9%
	Whole pancreas	1	2.2%
	Head	23	50.0%
	Head/Neck	3	6.5%
	Tail	3	6.5%
**Mass/duct abnormality at diagnosis**	Mass	26	56.5%
	Duct abnormality	11	23.9%
	Both	8	17.4%
	Missing site information	1	2.2%
**Basis of diagnosis**	Histology	21	45.7%
	Imaging alone	25	54.3%
**Mode of index imaging request**	Outpatient routine	9	19.6%
	Outpatient urgent	21	45.7%
	Outpatient surveillance	2	4.3%
	Inpatient urgent	14	30.4%
**Indication for index imaging**	Alarm abdominal symptoms	19	32.6%
	Non-alarm abdominal symptoms	10	17.4%
	Other indications for imaging	17	50%
**Modality of index imaging**	CT Abdomen and pelvis	16	34.8%
	CT Chest, Abdomen and Pelvis	14	30.4%
	CT Pancreas	3	6.5%
	CT Thorax	6	13%
	CT Colon	2	4.3%
	CT Urogram	1	2.2%
	CT Renal (multiphase)	1	2.2%
	MRI Abdomen	1	2.2%
	MRI Liver	1	2.2%
	MRI Renal	1	2.2%
**Total scans**	CT	43	93.5%
	MRI	3	6.5%
**Reporting radiologist**	Hepatobiliary or Gastrointestinal radiologist	15	32.6%
	Non-hepatobiliary or gastrointestinal radiologist	31	67.4%

MRI—Magnetic Resonance Imaging.

**Table 2 cancers-17-03770-t002:** The demographic details and diagnostic imaging characteristics of Post Imaging Pancreatic Cancer and Pancreatic Cancer patients at Sandwell and West Birmingham NHS Trust.

			PIPC Patients	%	PC Patients Without PIPC	%	*p*-Value
**Total**			23		263		
**Location of pancreatic cancer**		Body	7	30.4	62	23.6	
		Body and Tail	2	8.7	12	4.6	
		Head	13	56.5	117	44.1	
		Neck	1	4.4	7	2.7	
		Tail	0	0	42	15.9	
		Uncinate process	0	0	19	6.8	
		Missing site information			4	2.3	0.25
**Sex**		Male	10	43.5	138	52.5	
		Female	13	56.5	125	47.5	0.41
**Age at index imaging or diagnosis for pancreatic cancer without PIPC**			75.9	(63.4–80.2)	75.3	(65.1–81.4)	0.94
**Indication for imaging**		Alarm Abdominal symptoms	12	72.7	177	67.3	
		Non-alarm Abdominal symptoms	5	13.6	36	13.7	
		Other indications for imaging	6	13.6	50	19.0	0.87
**Reporting radiologist**		Hepatobiliary or Gastrointestinal radiologist	5	21.7	63	23.9	
		Non-hepatobiliary or gastrointestinal radiologist	18	78.3	198	75.3	
		Missing	0	0.0	2	0.8	0.89
**Diagnosis**		Pancreatic Ductal Adenocarcinoma	14	60.9	75	28.5	
		Pancreatic cancer diagnosed on imaging only	9	39.1	176	66.9	
		Neuroendocrine tumour	0	0.0	11	4.2	
		Small cell cancer	0	0.0	1	0.4	0.01
**Basis of diagnosis**		Histology	14	60.9	87	33.1	
		Imaging	9	39.1	176	66.9	0.01

PIPC—Post Imaging Pancreatic Cancer.

**Table 3 cancers-17-03770-t003:** Clinical staging and treatment received by pancreatic cancer patients with and without PIPC at Sandwell and West Birmingham NHS Trust.

		Pancreatic Cancer Without PIPC	%	PIPC	%	*p*-Value
**Total**	286	263		23		
**Clinical stage at diagnosis**						
**Early**	I	38	14.4	2	8.7	0.90
	II	44	16.7	3	13.0	
**Late**	III	37	14.1	4	17.4	
	IV	143	54.4	14	60.9	
**Not available**		1	0.4			
**Treatment intention**						
**Curative**	Surgical resection	31	11.8	3	13.0	
	Adjuvant chemotherapy	19	7.2	3	13.0	0.47
**Palliative**	Chemotherapy	42	16.0	3	13.0	
	Best supportive care	162	61.6	14	60.9	0.7
	Died before further management	23	8.7	1	4.3	

PIPC—Post Imaging Pancreatic Cancer.

## Data Availability

Reasonable requests for access to anonymized data may be considered by the corresponding author and will require review and approval by the participating NHS Trusts in accordance with local data protection regulations.

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
