# Peer review of "Root Cause Analysis of Patients with Pancreatic Cancer Who Underwent Imaging Not Resulting in a Cancer Diagnosis in the 18 Months Prior to Diagnosis"

_cancers, 2025, doi:10.3390/cancers17233770_

Round 1
Reviewer 1 Report
Comments and Suggestions for Authors
In this manuscript titled "A novel root cause analysis system to establish the most
plausible explanation for post imaging pancreatic cancer," the authors have presented study of pancreatic cancer patients whose diagnosis was missed at earlier radiological imaging (3-18 months) prior to real diagnosis. I do agree that this study is novel. I have following suggestions.
- I feel that the title of the paper does not actually convey what exactly that study wants to convey.
- They definitely need to make changes in the introduction with more plausible explanation what previous studies till now have emphasized. As till now, most of the studies have highlighted on lack of radiological criteria to predict response to neoadjuvant therapy.
- In the discussion section no where, they have mentioned what is the root cause and what guidelines should be provided.
- In the limitations section, only the point mentioning about the radiologists bias is clear. Further, when they mentioned that some patients may have been diagnosed with PIPC at different providers. It indicates that the information was not clearly available (or they want to emphasize on the shortage of number of cases).
Comments on the Quality of English Language
I request the authors to please review the sentences because sentences are not clear. They need to be short and clear.
Author Response
- I feel that the title of the paper does not actually convey what exactly that study wants to convey.
Response
We have altered the title to confer precisely what the study involved: Root cause analysis of patients with pancreatic cancer who underwent imaging not resulting in a cancer diagnosis in the 18 months before their diagnosis.
Text amended.
- They definitely need to make changes in the introduction with more plausible explanation what previous studies till now have emphasized. As till now, most of the studies have highlighted on lack of radiological criteria to predict response to neoadjuvant therapy.
Response
Thank you for this comment. The aim of this study is to establish why pancreatic cancer was not diagnosed in patients who previously had CT or MRI scan within 3-18 months of a scan that confirmed their diagnosis, through a structured root cause analysis. Therefore, radiological criteria to predict response to neoadjuvant therapy mentioned by the reviewer, does not fall within the remit of this manuscript. The studies quoted in the introduction by Ahn et al, Gangi et al and Kang et al looked at radiological features on pre-diagnostic CT imaging that were associated with subsequent pancreatic cancer diagnosis. A further study by Kang et al identified factors that may have confounded detection or interpretation of pancreatic cancer on prior imaging and found them to be: inherent tumor factors, concurrent pancreatic pathology, technical limitation and cognitive biases. The introduction has been changed to reflect findings from previous studies as suggested.
Text amended to introduction as suggested.
- In the discussion section no where, they have mentioned what is the root cause and what guidelines should be provided. Gangi et al’s study found that 50% of pancreatic cancers were visible in scans obtained 2-18 months prior to diagnosis and highlighted that pancreatic duct dilation and cut-off were early features of pancreatic cancer. However, both studies didn’t provide an explanation as to why pancreatic cancer was missed on pre-diagnostic imaging.
Response
Thank you for this comment. Unfortunately, there are no guidelines regarding the processes required when retrospectively assessing if a pancreatic cancer was potentially missed on previous imaging and whether this was potentially avoidable. The only study that provided potential explanation for pancreatic cancer missed on We have added more detail on the root cause analysis of PIPC to the discussion section as suggested by the reviewer.
Text amended.
- In the limitations section, only the point mentioning about the radiologists bias is clear. Further, when they mentioned that some patients may have been diagnosed with PIPC at different providers. It indicates that the information was not clearly available (or they want to emphasize on the shortage of number of cases).
Response
Thank you for this comment. This phenomenon is seen in studies of post colonoscopy colorectal cancer and post endoscopy upper gastrointestinal cancer of patients undergoing endoscopy in one provider and being diagnosed with cancer at another unrelated provider in about 10% of cases.
Text amended on PIPC to clarify.
Comments on the Quality of English Language
I request the authors to please review the sentences because sentences are not clear. They need to be short and clear.
Thank you.
Reviewer 2 Report
Comments and Suggestions for Authors
The study by Umar et al is interesting and can be beneficial for designing targeted strategies for pancreatic cancer early identification based on imaging techniques.
Two suggestions to improve the manuscript:
- Figure 1 requires a better representation.
- A schematic diagram can be included to summarize the overall findings of the study and future directions.
- Some minor typographical errors are present within the manuscript which requires attention.
Author Response
Two suggestions to improve the manuscript:
- Figure 1 requires a better representation.
Thank you for this comment. Figure 1 has been re-designed.
Response
Figure 1 amended on page 22.
- A schematic diagram can be included to summarize the overall findings of the study and future directions.
Thank you for this comment. Table 3 has been replaced by Figure 2 to depict the most plausible post-imaging pancreatic cancer explanations, but we can not include future directions in this figure. We are currently setting up a national study of PIPC in all hospitals in England and have added this to the discussion page 16
Response
Figure amended.
- Some minor typographical errors are present within the manuscript which requires attention.
Thank you. These have been addressed where found.
Reviewer 3 Report
Comments and Suggestions for Authors
Strong introduction which states the purpose of choosing this topic and justifies the importance of careful analysis of imaging in diagnosing pancreatic cancer . Good use of images throughout, as it provides a visual representation of the points mentioned throughout the article and provides a deeper understanding of pancreatic imagery to the readers who may not be as familiar with it . The authors concluded from their research that 3 out of 100 patients analysed had PIPC (post imaging pancreatic cancer) which backs up the importance of early diagnosis and analysis through imaging studies in patients with a suspicion of pancreatic cancer . Strong article with good english language use.
Author Response
Strong introduction which states the purpose of choosing this topic and justifies the importance of careful analysis of imaging in diagnosing pancreatic cancer . Good use of images throughout, as it provides a visual representation of the points mentioned throughout the article and provides a deeper understanding of pancreatic imagery to the readers who may not be as familiar with it . The authors concluded from their research that 3 out of 100 patients analysed had PIPC (post imaging pancreatic cancer) which backs up the importance of early diagnosis and analysis through imaging studies in patients with a suspicion of pancreatic cancer . Strong article with good english language use.
Response
Thank you for these encouraging comments.
Reviewer 4 Report
Comments and Suggestions for Authors
Early diagnosis is very important for radical treatment of pancreatic cancer. This retrospectively analysis of patients with pancreatic cancer previously undiagnosed on imaging found that 7.7% of patients were classified as having PIPC, of which 35% of PIPC were considered potentially avoidable.
Therefore, this study is highly significant. However, some concerns remain.
- Should the data in Tables 2 and 4 be indicated as simple numbers or percentages?
- Why are only 286 patients included in Tables 2 and 4? Why are the patients from LTHNHST (Leeds Teaching Hospitals National Health Service Trust) not included?
- The 3-month lower limit excludes cancers that became visible < 3 months after a scan; these are still clinically important“missed” cases. The 18-month upper limit is arbitrary; what if a tumor is diagnosed 19 months after a negative scan?
- Pre-diagnostic scans were reviewed by two radiologists “independently”, but the paper does not state whether they were blinded to later imaging or final diagnosis. Expectation bias may inflate the proportion of lesions retrospectively labelled“visible”.
- No confidence interval is given for either proportion (e.g., 7.7 % PIPC…)
- Language: “PIPC were …” should read“PIPCs were …” or “PIPC was”
Author Response
Early diagnosis is very important for radical treatment of pancreatic cancer. This retrospectively analysis of patients with pancreatic cancer previously undiagnosed on imaging found that 7.7% of patients were classified as having PIPC, of which 35% of PIPC were considered potentially avoidable.
Therefore, this study is highly significant. However, some concerns remain.
- Should the data in Tables 2 and 4 be indicated as simple numbers or percentages?
Response
Thank you for this comment. Table 4 is now Table 3. Both Tables 2 and 3 include both numbers and percentages and we think this is the best way to show the data.
Text not amended.
- Why are only 286 patients included in Tables 2 and 4? Why are the patients from LTHNHST (Leeds Teaching Hospitals National Health Service Trust) not included?
Response
Thank you for this question. Electronic records at Sandwell West Birmingham NHS Trust enabled ease of access to historical records for the research team, but unfortunately the same access was not possible at Leeds Teaching Hospitals NHS Trust
Text not amended.
- The 3-month lower limit excludes cancers that became visible < 3 months after a scan; these are still clinically important “missed” cases. The 18-month upper limit is arbitrary; what if a tumor is diagnosed 19 months after a negative scan?
Response
Thank you for this question. The 3-month lower limit was chosen because cancers diagnosed within 3 months of a CT or MRI scan are likely to have been present at the time and therefore represent a true positive result rather than a delayed diagnosis. It is recognised that there can be delays of a few months in pancreatic cancer due to the need for histological confirmation by endoscopic ultrasound. There are a limited number of studies which reviewed post-imaging pancreatic cancer, and the upper limit was 18 months in each of these. We have therefore adopted this threshold so that our study could be comparable with previous studies. We did comment in the results section that “Three patients had imaging more than 18 months before diagnosis and the duct dilatation on their PIPC imaging was a new finding in each patient.” To give some information on this.
Text not amended.
- Pre-diagnostic scans were reviewed by two radiologists “independently”, but the paper does not state whether they were blinded to later imaging or final diagnosis. Expectation bias may inflate the proportion of lesions retrospectively labelled“visible”.
Response
Thank you for this comment. We agree and in the limitation section we have explained that the radiologists were aware of the later pancreatic cancer diagnosis and that this awareness may have produced as the reviewer rightly states, an expectation bias which can overestimate the number of subtle lesions on non-diagnostic imaging.
Text amended to include referral to ‘expectation bias’ in the discussion section.
- No confidence interval is given for either proportion (e.g., 7.7 % PIPC…)
Response
Thank you. Text amended in the results section to include this on page 11 and 12.
- Language: “PIPC were …” should read“PIPCs were …” or “PIPC was”
Response
Thank you for this comment.
Text amended throughout the manuscript.
Round 2
Reviewer 2 Report
Comments and Suggestions for Authors
The revised manuscript looks better.